# Social support correlates with glucocorticoid concentrations in wild African elephant orphans

J. M. Parker [1,2,3✉], J. L. Brown[4], N. T. Hobbs[1,5], N. P. Boisseau[4], D. Letitiya[3], I. Douglas-Hamilton[3,6] & G. Wittemyer[1,2,3]

Social relationships have physiological impacts. Here, we investigate whether loss of the mother/offspring relationship has lasting effects on fecal glucocorticoid metabolite (fGCM) concentrations in wild African elephant orphans several years following their mothers' deaths. We find no difference in fGCM concentrations between orphans and nonorphans, but find lower fGCM concentrations in elephants with more age mates in their family. We also unexpectedly identify lower concentrations in orphans without their natal family versus nonorphans and natal orphans, which we speculate may be due to the development of hypocortisolism following a prolonged period without familial support. An index of plant productivity (i.e. food) shows the largest correlation with fGCM concentrations. Our findings indicate no lasting differences in glucocorticoid concentrations of surviving orphan elephants who are with their family, suggest the presence of age mates may reduce glucocorticoid concentrations in elephants, and emphasize that basic survival needs are the primary regulators of the stress response.

[1] Graduate Degree Program in Ecology, Colorado State University, 102 Johnson Hall, Fort Collins, CO 80523, USA. [2] Department of Fish, Wildlife and Conservation Biology, Colorado State University, 1474 Campus Delivery, Fort Collins, CO 80523, USA. [3] Save the Elephants, Marula Manor, Marula Lane, Karen, Nairobi 00200, Kenya. [4] Center for Species Survival, Smithsonian Conservation Biology Institute, 1500 Remount Road, Front Royal, VA 22630, USA. [5] Natural Resource Ecology Laboratory, Colorado State University, Fort Collins, CO 80523, USA. [6] Department of Zoology, Oxford University, Oxford OX1 3PS, UK. ✉email: jeparker@sdzwa.org

Studies have shown that supportive relationships with other members of the same species diminish an individual's stress response. This phenomenon is referred to as "social buffering"[1,2]. When a vertebrate is confronted with a stressor, the adrenal glands release more glucocorticoid (GC) hormones into the bloodstream. Social buffering is demonstrated when the presence of one or more companions reduces such GC release. For instance, strong bonds with other males lessened GC release in wild male macaques (*Macaca sylvanus*)[3], and the presence of a familiar companion lessened GC release following exposure to a stressor in captive male squirrel monkeys (*Saimiri sciureus*)[4]. While most studies of social buffering have concerned primates and laboratory rodents, it has been observed in additional taxa including fish and birds[1].

There is evidence that the mother-offspring bond plays a particularly important role in buffering the stress response. Moreover, maternal care early in life seems to program lifelong responses to stress[5–7]. Laboratory Norway rats (*Rattus norvegicus*) who received more licking and grooming from their mother in the first ten days of life released lower amounts of GCs when confronted with acute stressors later in life[8]. They also developed more GC receptors in the brain, enhancing the body's ability to stop secreting GCs after sufficient amounts have been released and thus reducing the chance of prolonged exposure to GCs that can cause health problems[7–12]. Additionally, zebra finch (*Taeniopygia guttata*) adults who were deprived of maternal care as chicks secreted more GCs when faced with social isolation than adults raised by both parents[13], and the GC concentrations of weaned guinea pigs (*Cavia porcella*) placed in a novel environment increased less if their mother was present than if a sibling or other conspecific was present[14,15]. If maternal care is lost, non-parental familial relationships can mitigate the stress response associated with that loss. For instance, an aunt or other alloparental figure lessened GC increases in infant squirrel monkeys temporarily separated from their mother[16].

Our current understanding of maternal effects on adrenal activity and the potential for other relationships to offset stress responses associated with maternal loss is derived mostly from captive studies, but there have been two recent studies on this topic in wild primates. Wild chimpanzee (*Pan troglodytes*) orphans showed short- but not long-term increases in glucocorticoid concentrations following their mother's death. Adoption occurs in chimpanzees and may ameliorate initial increases in GC concentrations following maternal loss[17]. Baboon (*Papio cynocephalus* and *Papio Anubis*) orphans had weaker social ties with other females following the loss of their mother but did not show definite differences in GC concentrations from nonorphans unless they had also faced other early-life adversity in addition to maternal loss[18]. These two studies provide evidence of a limited and context-dependent effect of maternal loss on GC levels in wild primate individuals.

Like chimpanzees and baboons, African elephants (*Loxodonta africana*) are a long-lived, highly social species with a high encephalization quotient that indicates advanced cognitive abilities[19–21]. They offer an additional opportunity to measure how GC concentrations relate to maternal care and social ties in a wild setting and to expand such studies to a non-primate taxon. The mother-calf bond is fundamental to the well-being of elephant calves, even post-weaning and into young adulthood[22–25]. Bonds among individuals are strong; females usually remain with their family throughout life in matriarchal core groups, and close familial bonds have been associated with lower GC concentrations[26–28]. Further, African elephants are endangered due to habitat loss and poaching[29–31]. The latter removes maternal and other adult caregivers from families, therefore understanding elephants' stress response to the loss of fundamental bonds can inform conservation[31,32].

The population of wild elephants that use the Samburu and Buffalo Springs National Reserves of Kenya (hereafter the "Samburu population") has been studied for over 20 years[31,33]. Poaching and severe drought from the years 2009–2014 killed many adult females in the Samburu population, leaving behind fragmented families and known female orphan calves[27,31]. These orphans have been the focus of several studies, therefore we know they survive less than nonorphans even if they are weaned at the time of their mother's death, and that orphans who leave their families to join other families or group together with other orphans suffer increased aggression and are isolated from adult females[22,23,25]. In this study, we investigated whether the loss of maternal care correlates with long-term increases in GC secretion in wild African elephant orphans and whether the presence of other family members may dampen changes in secretion. We defined "orphan" as a female whose mother died before that female reached maturity, with maturity marked by giving birth. We preferentially sampled from orphan subjects that had been part of earlier studies, resulting in a mean time since mother's death among orphan study subjects of 5.54 years (± standard deviation of 2.67 years, Supplementary Fig. 1F). We hypothesized maternal presence attenuates adrenal activity in wild African elephants and given the strong lifelong mother/offspring bond and lack of adoption in elephant society, we predicted orphans would sustain higher fecal glucocorticoid metabolite (fGCM) concentrations than nonorphans even years after their mother's death. We also hypothesized that the presence of social partners dampens GC secretion, predicting that orphans and nonorphans with more adult caregivers and age mates in their core group would have lower fGCM concentrations.

## Results

**Fecal glucocorticoid metabolite concentrations**. The dry weight average fGCM concentration across $n = 496$ dung samples from 37 elephants ($n = 25$ orphans, $n = 12$ nonorphans, Fig. 1) who ranged in age from 7–21 years (mean $13.36 \pm 3.09$ years) was $95.69 \pm$ SD $30.21$ ng/g, from minimum 14.19 ng/g to maximum 246.56 ng/g. We included time series graphs of individual study subjects' concentrations in Supplementary Fig. 2, as there is a call for information on the repeatability of wild individuals' GC concentrations in the literature[34,35].

**Individual-based variables**. We structured our Bayesian hierarchical model of fGCM concentrations into two levels, the first to estimate the effects of individual-based variables on baseline GC secretion and the second to estimate the effects of variables that change with time on GC secretion at the time of sample collection. For the first level of the model, three of the five individual-based variables assessed had confidence intervals that did not overlap 0 (i.e., these three variables were statistically significant): if the individual remained with her natal family, the number of age mates in the core group, and the number of samples collected from an individual (Table 1 and Fig. 2). Orphan individuals who left their natal family to join another family or group together with other orphans showed lower average fGCM concentrations than orphans who remained with their natal family and nonorphans. Additionally, individuals with more age mates in their core group had lower average fGCM concentrations than individuals with fewer age mates, although the magnitude of this correlation was not as large as that of being with a non-natal family. Finally, more samples collected from an individual correlated with lower average fGCM concentrations. The model estimated that more adult females in a core group is associated with slightly lower

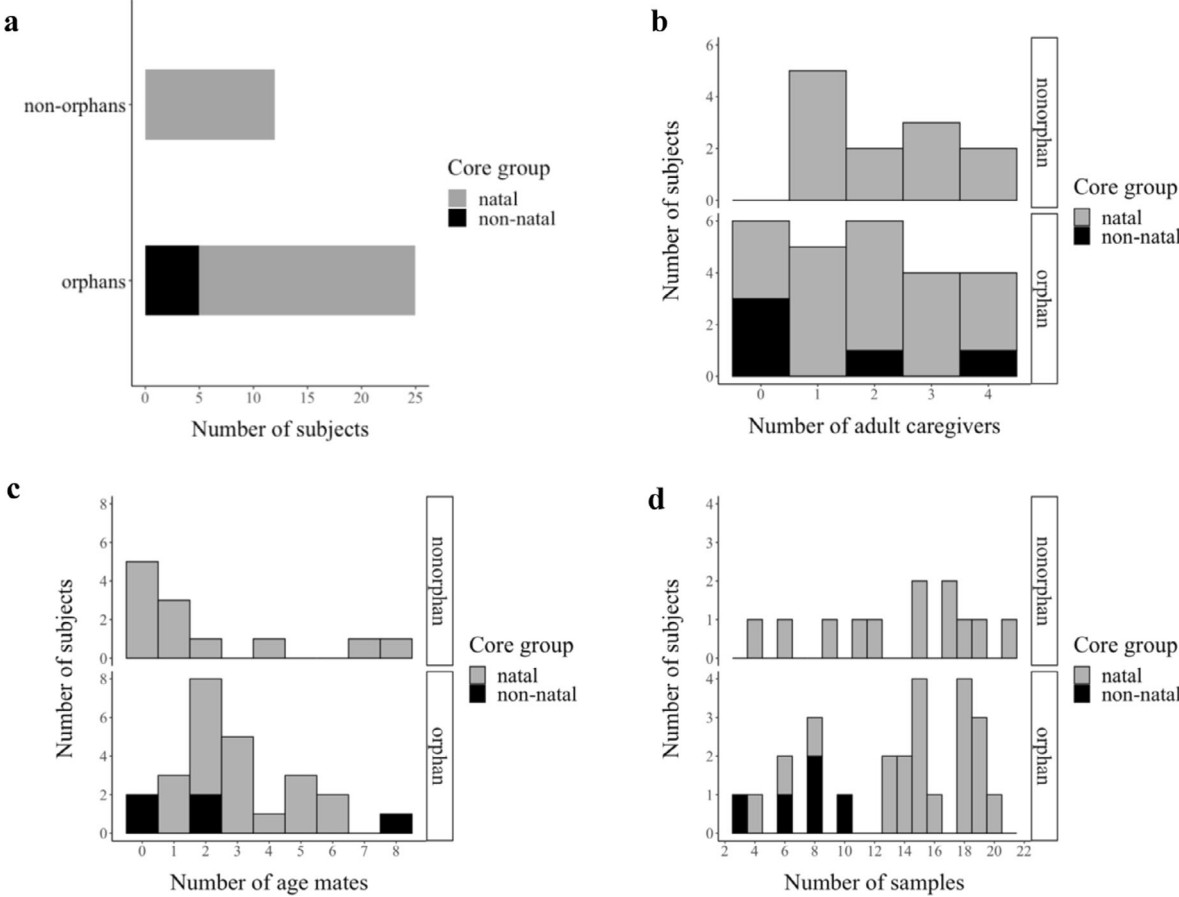

**Fig. 1 Summary of sample size distributed according to orphan status, family group composition, and dung samples per individual. a** Bar chart showing the number of study subjects who were orphans ($n = 25$ elephants) versus nonorphans ($n = 12$ elephants), colored according to whether they were with their natal ($n = 32$ elephants, shaded gray) versus non-natal ($n = 5$ elephants, shaded black) core group. **b–d** Histogram of study subjects according to the number of adult female caregivers in their core group, the number of age mates in their core group, and the number of dung samples, each also separated into orphan versus nonorphan panels and colored according to natal versus non-natal core group. Note that the nine discarded dung samples (see methods) are not included in **d**.

fGCM concentrations, but the 95% confidence interval for this estimate overlapped zero. Orphan status showed no correlation with fGCM concentrations.

**Time-variant variables**. For the second level of our model, three of the eight time-varying variables assessed had confidence intervals that did not overlap 0: the mean normalized difference vegetation index (NDVI—a measure of primary productivity/greenness of vegetation across the landscape), a standard deviation of NDVI (a measure of how predictably distributed resources were across the landscape), and the amount of time a sample spent on the ground prior to collection (Table 1 and Fig. 2). Higher mean NDVI, signaling more available food resources, strongly correlated with lower fGCM concentrations. Higher NDVI standard deviation, signaling patchily distributed resources, strongly correlated with higher fGCM concentrations (Fig. 3). The amount of time a sample spent on the ground before being collected into a cool box correlated with slightly higher fGCM concentrations. The 95% confidence intervals for all other variables overlapped zero, but lactating and older age showed a weak negative correlation with fGCM concentrations while spending more years without a mother and pregnancy showed a weak positive correlation with fGCM concentrations. Time of day was not correlated with fGCM concentrations.

**Strongylid fecal egg counts**. We conducted a second analysis to assess the correlation between fGCM concentrations and strongylid (a type of gastrointestinal parasitic worm) fecal egg counts (FECs) using the same model structure, but with a smaller sample size ($n = 446$) due to fewer samples with matched FEC data. fGCM concentrations showed a weak positive correlation with FECs, but the 95% confidence interval overlapped zero (Supplementary Table 1 and Supplementary Fig. 3).

**Model checks**. Posterior predictive checks[36] showed no evidence of lack of fit. The Bayesian $p$ value for the mean of the first level of the model (estimating individual baseline fGCM concentrations) was 0.51, and 0.28 for the standard deviation. The Bayesian $p$ value for the mean of the second level (estimating the effect of time-variant covariates) was 0.50 and 0.74 for the standard deviation. Corresponding Bayesian $p$ values for the model with fewer samples testing for a correlation with strongylid fecal egg counts were 0.36, 0.28, 0.54, and 0.76.

## Discussion
Our study investigated how maternal loss affects adrenal activity in wild orphans of a long-lived, non-primate mammal species. Contrary to our expectation, we did not find evidence that surviving orphans sustained higher baseline fGCM concentrations several years following their mother's death, even though

elephant orphans suffer lower survival than nonorphans[25]. However, our second prediction was partially upheld because fGCM concentrations were lower for individuals in core groups with more age mates. This supports our hypothesis that social buffering occurs in wild elephants. However, our study was cor-

**Table 1 Bayesian hierarchical regression model results, showing estimated coefficients relating variables to fecal glucocorticoid metabolite concentrations, from 37 sampled elephants ($n = 496$ dung samples).**

| Coefficient | Covariate | Estimate | 95% CI lower | 95% CI upper |
|---|---|---|---|---|
| *$\beta_5$ | *with non-natal group | −0.41 | −0.81 | −0.01 |
| *$\beta_2$ | *age mates | −0.25 | −0.37 | −0.13 |
| *$\beta_3$ | *number of samples | −0.14 | −0.24 | −0.03 |
| $\beta_1$ | adult caregivers | −0.09 | −0.20 | 0.01 |
| $\beta_4$ | orphan status | 0.06 | −0.24 | 0.36 |
| *$\gamma_5$ | *mean NDVI | −0.69 | −1.07 | −0.31 |
| $\gamma_2$ | lactating | −0.10 | −0.35 | 0.15 |
| $\gamma_1$ | age | −0.06 | −0.21 | 0.08 |
| $\gamma_4$ | time of day | 0.02 | −0.07 | 0.11 |
| $\gamma_8$ | years without mom | 0.07 | −0.07 | 0.22 |
| $\gamma_6$ | pregnancy | 0.08 | −0.12 | 0.29 |
| *$\gamma_3$ | *time sample sat on ground | 0.09 | +0.00 | 0.18 |
| *$\gamma_7$ | *NDVI standard deviation | 0.79 | 0.41 | 1.18 |

The extra line space separates the first (above line) and second (below line) levels of the model. The first level estimated individual-level intercepts, determining the correlation of unchanging variables with mean fecal glucocorticoid metabolite concentrations. The second level used the estimated intercepts of the first level to determine the correlation of variables that changed with time according to when a sample was collected. Coefficients and covariates are ordered according to the estimated effect size, from negative to positive.
*NDVI* normalized difference vegetation index, a measure of primary productivity.
*denote where the estimated 95% confidence interval did not overlap zero.

relational, therefore we cannot assert with certainty that the presence of age mates attenuated the stress response and caused the observed lower fGCM concentrations. We did not observe a significant correlation with number of adult caregivers as predicted. Goldenberg and Wittemyer (2017) found that orphan elephants interact more with age mates and bulls than do nonorphans, and one reason for strengthening such social bonds may be that they reduce GC concentrations[22]. Conversely, orphan elephants interact less with adult females in their core group than do nonorphans[22]. Therefore female presence might be less likely to lower GC concentrations for orphans, who made up the majority of our study subjects. Relatively young elephants like those included in this study may also be displaced by older adult females while feeding or resting, as age is strongly correlated with social rank in African elephants[22,37], and this could be a counterbalancing source of increased GC secretion from the company of adults. Overall, if the lower fGCM concentrations associated with age mates indicates that social buffering occurs in African elephants, that buffering seems to be dependent on the age and relatedness (see below) of available conspecifics.

Surprisingly, orphans who were no longer with their natal group, of which a majority came from families with no adults left due to poaching, demonstrated lower average fGCM concentrations than nonorphans and orphans who remained with their families. Given the challenges experienced by these orphans, including receiving more aggression from other individuals[23], we surmised they would show the greatest fGCM concentrations among the study subjects because increased secretion of GCs is traditionally equated with greater stress. We speculate our opposite results may be because the stress response is complicated and adrenal glands can exhibit either hyper- or hypo-secretion depending on the type and duration of a stressor[7,38–40]. Long-term social stressors, especially those whose onset is during developmental stages, can cause prolonged hypersecretion of GCs that eventually down-regulate adrenal activity, which could be an adaptive response that prevents harmful effects of consistently elevated GC concentrations[41,42]. This downregulation results in sustained lower baseline GC concentrations, a phenomenon

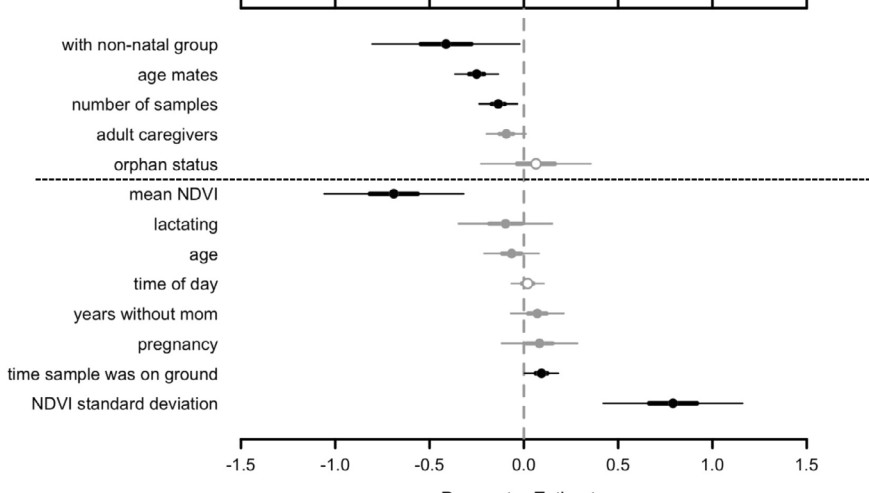

**Fig. 2 Coefficient values and 95% confidence intervals from Bayesian hierarchical regression model ($n = 496$ dung samples from 37 elephants).** Black denotes estimates whose 95% confidence interval did not overlap zero, gray denotes estimates whose 50% confidence interval did not overlap zero, and open circles denote variables for which both confidence intervals overlap zero. The dotted line separates the first (above line) and second (below line) level of the model. The first level estimated individual-level intercepts, determining the correlation of unchanging variables with mean fecal glucocorticoid metabolite concentrations. The second level used the estimated individual intercepts of the first level to determine the correlation of variables that changed with time according to when a sample was collected. Covariates are ordered according to estimated correlation size, from negative to positive, within each level. NDVI stands for "normalized difference vegetation index," a measure of primary productivity.

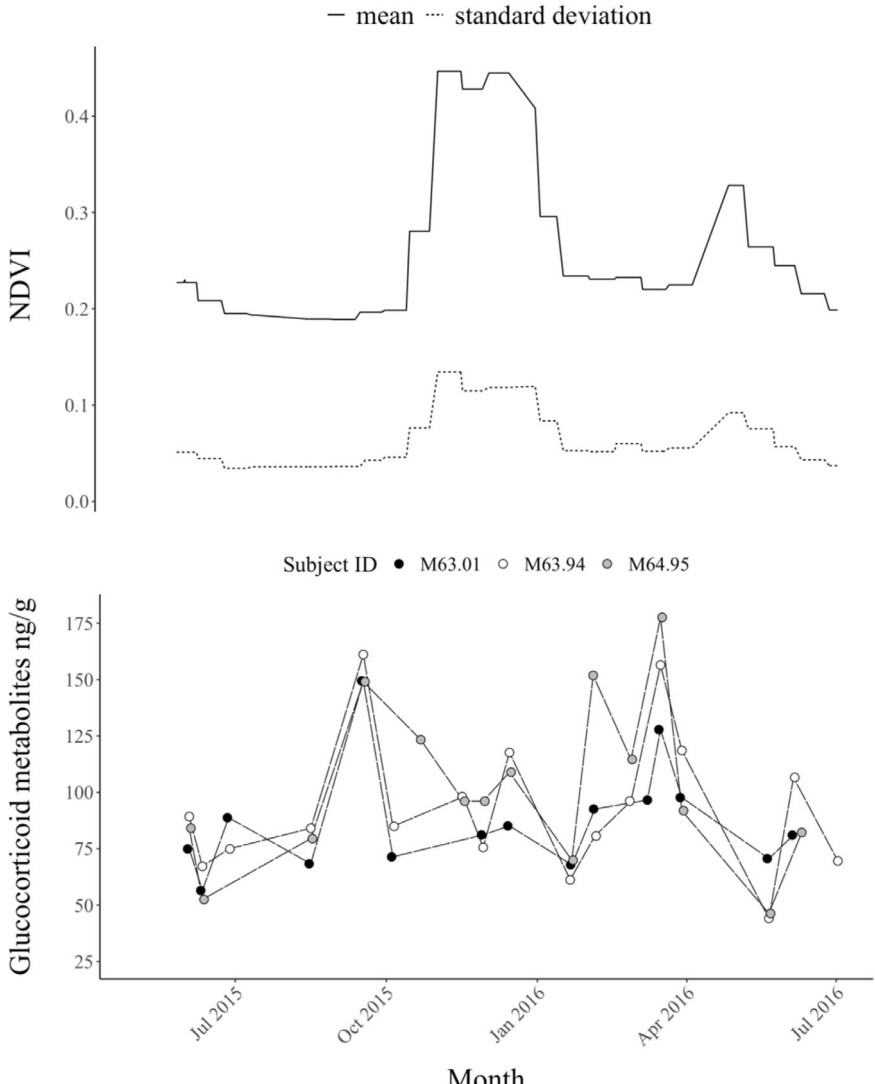

**Fig. 3 Depiction of the strong correlation between fecal glucocorticoid metabolite concentrations and seasonality.** NDVI (normalized difference vegetation index—a measure of primary productivity) mean (solid line) and standard deviation (dotted line) over the course of the study above the glucocorticoid metabolite concentrations in ng/g (nanograms of metabolites per gram of fecal matter) of three study elephants from the same family. Circles depict sample fecal glucocorticoid metabolite concentrations, with samples from each elephant distinguished according to fill color (white, black, or gray). Long-dashed lines emphasize oscillations in a single elephant's concentrations over time. Glucocorticoids fluctuated most in relation to changes in primary productivity.

termed hypocortisolism that has been associated with physiological problems such as autoimmunity, depression, and excessive fatigue in humans (Homo sapiens); skin and oral lesions in black rhinos (Diceros bicornis); and depressive behavior, vertebral problems and hematological anomalies like anemia in horses (Equus caballus)[38,40,43–47]. Supposing the absence of family initially causes a prolonged rise in GC secretion for non-natal orphans, we are potentially observing the results of an eventual downregulation of the adrenals over time given we sampled several years after the death of orphans' mothers. Hypocortisolism in non-natal orphan elephants would be in line with some other studies of cognitively advanced mammals. For example, a study in captive rhesus monkeys (Macatta mulatta) found that maternal loss was associated with hypocortisolism; monkeys separated from their mothers at birth had significantly lower basal cortisol concentrations compared to controls even three years following separation[48]. Perhaps more relevant given a lack of social support for non-natal orphans, a study in humans found that AIDS-orphaned children in South Africa who perceived a high degree of social support from siblings, caretakers, friends, and others were less likely than their peers to develop post-traumatic stress disorder, a common symptom of which is hypocortisolism[38,49,50].

We designed this study to assess the effect of social context on glucocorticoid secretion, but it is clear that ecological factors related to resource availability are the primary driver of glucocorticoid secretion in wild African elephants. fGCM concentrations within a single individual were highly variable, oscillating according to season. Concentrations were lower when mean NDVI was higher and vice versa, indicating that when food was readily available the study subjects' GC secretion was lower and when food was of lower quality their GC secretion was higher. Conversely, fGCM concentrations were higher when NDVI standard deviation was high, which rises during seasonal transitions when rainfall is spatially and temporally stochastic, and presumably signals greater unpredictability in the distribution of prime food resources across the landscape. Thus, we suspect higher fGCM concentrations were related to food searching

behavior[51]. Basic survival needs are recognized as the primary regulators of the stress response[7,42], therefore the substantial correlation of fGCM concentrations with NDVI measures was expected[52]. The slight but significant correlation between the more time a sample spent on the ground and higher fGCM concentrations was potentially related to NDVI as well. During dry seasons elephants spend more time near the riverbed[53] in treed areas that made it difficult to drive up to samples, and we had to wait longer for them to move a sufficient distance away than when in open areas.

Although our study was robust in the use of longitudinal sampling from well-known subjects of a long-term monitoring project, we could not sample evenly from all subjects. This may have affected results, particularly given that more samples were collected from an individual slightly but significantly correlated with lower average fGCM concentrations. We do not know the reason for this correlation, but we collected fewer samples from non-natal orphans and still found they had lower concentrations than more heavily sampled individuals. Additionally, the sample size of orphan subjects who had left their natal family was small ($n = 5$), and they tended to be in contexts with fewer adults and age mates. We could not remedy this because few orphans in the study system left their natal groups[23]. (See supplementary discussion.)

Regarding conservation, our results are hopeful for elephant populations that still contain functional family units. They suggest orphaning does not have lasting effects on the adrenal activity of surviving African elephant orphans provided they remain with their natal group. The level of poaching in the Samburu population was not high enough to fragment the majority of core groups and most orphans stayed with their families. Where heavier poaching has fragmented a greater number of family units, non-natal orphans would be more common, and additional study to investigate whether lower fGCM concentrations develop over time in those populations would be elucidating[54]. Concerning management, evidence that social support can compensate for maternal loss is useful for caring for orphan elephants brought into captivity. Providing age mates and maintaining core groups of bonded orphans may reduce the stress associated with captivity[55], and releasing bonded groups together could ease their transition back into the wild[56].

While not possible in our study, it would be valuable to sample wild elephant orphans immediately following their mother's death to assess if there is an initial increase in GC secretion. Subsequent longitudinal sampling would then show if adrenal steroidogenic activity changes over time and if negative feedback may truly be the cause of the lower concentrations observed in non-natal orphans. Extended research is also needed to investigate whether there are long-term fitness effects associated with our findings in non-natal orphans. Importantly, we cannot equate altered GC concentrations with reduced fitness[57,58]; non-natal orphans in this study survived the duration of sampling, despite orphans having a lower survival probability in general[25].

Measuring the physiological responses of individuals to changes in their social environment helps us to understand the benefits of sociality. Through measuring fGCM concentrations of wild African elephant orphans, we observed correlations that point to the importance of age mates and familial relationships in buffering maternal loss for surviving orphans. Preserving social bonds within wildlife populations may make individuals within those populations more resilient to disturbance and optimize their physiological condition.

## Methods

**Study system and subjects.** The Samburu and Buffalo Springs National Reserves are unfenced reserves in Kenya, located at 0.3–0.8°N, 37–38°E and divided by a

semi-permanent river called the Ewaso Ngiro[33]. Together they encompass 220 km² of semi-arid terrain, with an annual average rainfall of 350 mm during two wet seasons from April–May and November–December, and elevations ranging from 800–1200 m[31,33]. African elephants who use the reserves are part of a long-term monitoring study that began in 1998, and to date over 1000 elephants have been identified and are followed daily, such that detailed demographic history is available for each individual[31].

For this study, we selected 37 female elephant subjects for longitudinal sampling. We chose age-matched (i.e., within 4 years of age, see below) orphan/nonorphan subjects based on how often they were in the reserves for ease of sample collection. We preferentially chose individuals who were part of earlier orphan studies (e.g., ref. [22]) to advance wholistic understanding of the consequences of being orphaned. The subjects ranged in age from 7–21 (mean $13.36 \pm 3.09$) years at the start of collection (see Supplementary Fig. 1B). Subjects were categorized as an orphan ($n = 25$) or nonorphan ($n = 12$), where an orphan was defined as a female whose mother died before that female reached maturity, with maturity marked by giving birth (Fig. 1a). The orphan subjects had lost their mothers from 1.76–19.04 (mean $5.54 \pm$ standard deviation 2.67) years prior to the start of our study, therefore we were testing for long-term GC alterations. Of the 25 orphans, five had left their natal family to join an unrelated core group or form a group with other orphans after their mother's death (Fig. 1a).

**Fecal sample collection.** We longitudinally sampled dung over a period of 13 months from June 2015 to July 2016, with a minimum of two weeks between collecting samples from the same individual. Upon finding a study subject in the field, we stayed with them until they produced a sample. Elephants in the Samburu population who frequent the reserves are habituated to and will feed or rest alongside vehicles[31]; therefore, we assumed our presence did not represent a stressor. When a sample was produced, we recorded the GPS and defecation time. Once the subject had moved away, we drove near to the sample (using our permission to go offroad only when necessary for safety and driving slowly to minimize damage), labeled a 30 mL plastic bottle with the subject ID and date, then filled the labeled bottle with homogenized dung from the center of at least two boluses[59], placed it in a cool box and recorded collection time. Often dung from the same boluses was also collected into a plastic bag for strongylid fecal egg counts. Upon returning to the research camp, within a maximum of eight hours following collection, samples were stored in a freezer at approximately −10 °C until analysis.

**Fecal extraction and fGCM analysis.** We shipped the resulting 505 fecal samples on dry ice to the Smithsonian Conservation Biology Institute in Front Royal, VA. There, they were lyophilized (VirTis from SP Industries, Warminster, PA) and crushed, then 0.1 g of the resulting powder was put in labeled $16 \times 125$ mm glass tubes (Fisher Scientific; Pittsburgh, PA) and 5 mL of 80% methanol was added to each tube. The tubes were capped with rubber stoppers, mixed on a multi-tube vortexer (Glas-Col; Terre Haunte, IN) for 30 min, and centrifuged for another 20 min at 2500 rpm (Sorvall RC 3 C Plus; Thermo Fisher Scientific, Waltham, MA). Supernatants were decanted from each sample into another set of labeled tubes, and the leftover pellets were suspended in 80% methanol, vortexed for 1 min, and centrifuged for 20 min at 2500 rpm. The two supernatants were combined and dried under air in a fume hood, then mixed with 1 ml of 100% methanol, dried again, and finally suspended in 1 ml of buffer (0.149 M NaCl, 0.1 M NaPO₄, pH 7.0). The tubes were sonicated (Part# 08895-60; Col-Parmer, Vernon Hills, IL) for 30 seconds to fragment and dissolve particles. Each extract was diluted (1:5) in enzyme immunoassay (EIA) buffer (Cat. No. X065, Arbor Assays, Ann Arbor, MI, USA) and stored at −20 °C before EIA analysis.

fGCM concentrations were measured with a double-antibody enzyme EIA containing polyclonal rabbit anti-corticosterone antibody (CJM006), validated for elephant fecal samples[60,61]. Samples (50 μL each) were added to pre-coated goat anti-rabbit IgG 96-well plates at room temperature, followed by addition of corticosterone-horseradish peroxidase (25 μL, 1:20,000 dilution) and anti-corticosterone antibody (25 μL, 1:60,000 dilution). Plates were covered with microplate sealers, incubated at room temperature on an agitator (Model E6121; Eberbach Corp., Ann Arbor, MI) for 1 h, and washed four times (1:20 dilution, 20X Wash Buffer Cat. No. X007; Arbor Assays) and blotted dry. Next 100 μL of TMB (3, 3′, 5, 5′-tetramethylbenzidine) (Moss Inc., Pasadena, MD) was added to each well, plates were incubated at room temperature for 30-45 min without agitation, then 50 μL of 1 N HCl solution was added to stop reactions. Optical density was read in a plate reader at 450 nm (OPsys MR; Dynex Technologies, Chantilly, VA). Assay sensitivity, based on 90% binding on the standard curve, was 0.14 ng/ml. Intra- and inter-assay CVs were <10 and 15%, respectively.

**Choice and calculation of covariates.** We used data on the number of adult caregivers and age mates within an individual's core group to test for social buffering effects. We defined "adult caregiver" as a multiparous female, given first-time mothers are still young and inexperienced (some as young as 9 years old[31]). The number of adult caregivers available in a subject's core group, $f_i$, ranged from 0 to 4 (Fig. 1B). We defined "age mate" as an undispersed male or female individual within 4 years of age because 4 years is the average interbirth interval for the Samburu population[31]. The number of age mates, $m_i$, available within a subject's

core group ranged from 0 to 8 (Fig. 1c). Orphan status, $o_i$, was represented as a 0/1 binary variable with 0 = nonorphan and 1 = orphan. Another binary variable, $a_i$, represented whether an individual was with a related core group, with 0 = with a natal family and 1 = with an unrelated core group.

Additionally, we included several covariates that were factors associated with GC production in previous elephant studies, namely reproductive condition (e.g., ref. [52]), age[62], time of day[63], and seasonality[62]. The long-term demographic monitoring data provided precise information on reproductive conditions and age. Binary 0/1 variables represented if a subject was pregnant, $p_{ij}$, and/or lactating, $l_{ij}$, versus not at each sampling event (Supplementary Fig. 1A). A subject's precise age at each sampling event, $g_{ij}$, was represented as a continuous variable (Supplementary Fig. 1B). We recorded the time of day a sample was produced as described above (Supplementary Fig. 1C), and represented it as a continuous variable, $t_j$, from 0 to 1, with 0 = midnight and 1 = 23:59:59. For seasonality, we used the normalized difference vegetation index (NDVI) 16-day composite images from the moderate resolution imaging spectroradiometer (MODIS) to estimate primary productivity during the collection period[64]. Rasters were clipped to a core study area drawn from elephant GPS collar data[53], then the mean, $v_j$, and standard deviation, $z_j$, of pixel values were extracted for each sampling date (Supplementary Fig. 1E). The mean was interpreted as an overall measure of productivity, and the standard deviation as a measure of vegetation predictability[51].

Additionally, we included as control variables the number of samples per individual, $n_i$ (mean ± SD: 15 ± 4 samples per individual, Fig. 1d) and the time between when a sample was produced and then placed in the cool box (Supplementary Fig. 1D), $s_j$, in case of degradation[65]. Further, since orphan subjects varied in the amount of time since their mother had died, we included the number of years since the mother's death at a time of collection, $b_{ij}$, as a continuous variable (Supplementary Fig. 1F).

Finally, as some studies have shown the effects of parasites on GC secretion in vertebrates[66], and because many of our samples were paired with fecal egg counts (FECs) to estimate strongylid infection for another study[67], we ran an analysis including those FECs, $w_{ij}$, as a covariate. The FECs (Supplementary Fig. 1G) were obtained as described in ref. [67] using the McMaster slide method[68]. Briefly, dung taken from the same boluses as the fGCM samples was mixed with saltwater to float strongylid eggs, then the solution was pipetted into slides with an overlain grid. The easily recognizable eggs falling within the grid were counted under a 10 ×10 microscope, and these counts entered in a standard formula to calculate the approximate number of strongylid eggs per gram of fecal matter[68].

**Statistics and reproducibility**. Prior to analysis, we discarded samples that were greater than three standard deviations from the mean fGCM concentration of the individual from which they were collected, as these outliers were likely misrepresentative and removing them greatly improved our model fit. In total, we discarded nine of 505 samples, each from a different individual, and were left with 496 samples for the analysis described below. (See Data Availability statement to view fecal glucocorticoid metabolite data and discarded samples.)

We analyzed the data using a hierarchical Bayesian model with uninformative priors (see Code Availability statement). One level of the model used time-invariant, individual-based covariates (defined above; Fig. 1) to estimate the mean baseline fGCM concentration for each individual:

$$\bar{y}_i \sim \text{Normal}\,(\mu_i, \tau_1) \qquad (1)$$

$$\mu_i = \alpha + \beta_1 f_i + \beta_2 m_i + \beta_3 n_i + \beta_4 \circ_i + \beta_5 a_i \qquad (2)$$

where $\bar{y}_i$ = mean fGCM concentration of individual $i$'s samples, distributed normally with parameters $\mu_i$ = the true unknown mean baseline fGCM concentration of individual $i$ and $\tau_1$ = a precision parameter associated with measurement uncertainty and uncertainty surrounding the process by which an individual's baseline fGCM values are determined.

The estimated mean baseline from the first level model (Eq. 2) was the intercept within the linear deterministic process model of the second level (Eq. 4), which estimated the effects of previously defined covariates that change with time (Supplementary Fig. 1):

$$y_{ij} \sim \text{Normal}\,(\mu_{ij}, \tau_2) \qquad (3)$$

$$\mu_{ij} = \mu_i + \gamma_1 g_{ij} + \gamma_2 l_{ij} + \gamma_3 s_j + \gamma_4 t_j + \gamma_5 v_j + \gamma_6 p_{ij} + \gamma_7 z_j + \gamma_8 b_{ij} \qquad (4)$$

where data $y_{ij}$, are distributed normally with parameters $\mu_{ij}$ = the true unknown fGCM concentration of the sample from individual $i$ at sampling event $j$, and $\tau_2$ = a precision parameter again associated with uncertainty. The structure of the model was such that the two levels informed one another and samples from a single individual were grouped together with an individual-level intercept.

We ran our analysis in RStudio version 1.1.463[69,70] using the package *rjags* version 4-10[71] with Markov-Chain Monte Carlo, running three parallel chains of 100,000 iterations, using 1000 for adaptation and discarding 10,000 as burn-in. We checked convergence with Gelman–Rubin diagnostic Rc values[72,73], which were all <1.1. All covariates and fGCM concentration data were standardized by subtracting the mean and dividing by the standard deviation ($\frac{x-\bar{x}}{\sigma}$) prior to analysis. We assessed model fit by calculating Bayesian $p$ values for the mean and standard deviation of each model level.

FEC counts were available for only 446 of the 496 samples, therefore we ran a separate analysis with the subset of 446 samples to test for a correlation with strongylid infection[67]. This analysis had the same format but with the addition of the term "$+ \gamma_9 w_{ij}$" ($w_{ij}$ described above).

**Figures**. We made Figs. 1, 3, and Supplementary Figs. 1, 2, in RStudio version 1.1.463 using ggplot2[69,70,74]. We made Fig. 2 and Supplementary Fig. 3, using the *MCMCvis* package[75].

**Reporting summary**. Further information on experimental design is available in the Nature Research Reporting Summary linked to this paper.

## Data availability
The fGCM concentration data are available at https://doi.org/10.7910/DVN/WM74G3. Due to the endangered conservation status of the African elephant, other data used in the manuscript are sensitive but may be viewable upon request. Some of the content in this publication was originally published in the first author's Ph.D. dissertation[76].

## Code availability
Our Rjags model specification code is available at https://doi.org/10.7910/DVN/NS0MJT. All other computer code used in analyses will be made available upon request.

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

## Acknowledgements

We thank the National Commission of Science and Technology (NACOSTI) of Kenya, the National Environment Management Authority (NEMA) of Kenya, the Kenya Wildlife Service (KWS), the county governments of Samburu and Isiolo, and the Convention on International Trade in Endangered Species (CITES) for allowing us to conduct this research in the Samburu and Buffalo Springs National Reserves and export fecal samples to the United States. Thank you to Save the Elephants for providing a research base, and further to D. Daballen, W. Kimani, F. Pope, T. Obari, F. Omengo, M. Mosse, L.W. Muita, and G.S. Sabinga for logistical support. S.Z. Goldenberg helped to initiate sample collection, and R. Kuriyan Wittemyer, N. Wittemyer, and K. Wittemyer assisted with sample collection. K. Crooks and L. Pejchar provided helpful comments on the manuscript, and the comments of three anonymous reviewers greatly improved its quality. This work was funded by the National Science Foundation Graduate Research Fellowship Program, grant number 006784, the Smithsonian Conservation Biology Institute, and S. Rankin of WorldWomenWork.

## Author contributions

J.M.P. formulated the research questions, conducted the literature review, designed the study, collected dung samples, performed the Bayesian analysis, and wrote the early draft and final version of the manuscript. J.L.B. led the laboratory analysis and contributed to writing the manuscript. N.T.H. reviewed the Bayesian analysis and contributed to writing the manuscript. N.P.B. coordinated the laboratory analysis and contributed to writing the manuscript. D.L. collected dung samples. I.D.-H. contributed to writing the manuscript. G.W. conceived the study, guided the formulation of the research questions, guided the design of the study, collected dung samples, and contributed to writing the manuscript.

## Competing interests

The authors declare no competing interests.
