## [Peer Review file · Communications Biology]

Reviewers' comments:

Reviewer #1 (Remarks to the Author):

The paper "Social support correlates with glucocorticoid concentrations in wild African elephant orphans", submitted by Parker and colleagues investigated the effect of maternal loss on fecal glucocorticoid metabolite (fGCM) concentrations in wild African elephants. In their interesting, longitudinal field study of well monitored African elephant groups in Kenya, the authors found lower fGCM levels in juveniles with more individuals in the core group, but no differences between orphans and non-orphans. Unexpectedly, orphans which had left their natal family had also lower fGCM concentrations.

I found it confusing that none of the stated (e.g. lines 5, and 7-8) and discussed (line 92 onwards) main findings can be explicitly found in the "Results" section, but only technical statements pointing to tables and figures. What about the statistical significance of those findings. In addition, some findings of the abstract (e.g. line 7) seem to contradict the title.

Nevertheless, I enjoyed reading and think the paper warrants publication. However, these and other points need to be clarified by the authors (see also detailed comments below).

Below, please find specific comments and questions (ordered by appearance in the ms):

Line 11: I don't think "resilience" can be concluded from your data and from a mere correlational study.

Line 13: "benefits of sociality": I guess the other findings (lines 5-6) may underline this)

Line 13: "and contributing to our understanding of why it evolved". I suggest deleting this part. It would be similar true in case of opposite results and what do we understand here now better?

Line 66: A minimum fGCM concentration of 10 pg/g seems very low. What was the detection limit of the method (in ng/g)? And by looking at the suppl Fig 2, this value seems to be an artefact (not reasonable – and there are a few other of those very low levels – please re-check). But most likely those were already removed (lines 292-294).

Line 72: NDVI. This abbreviation pops up here for the first time and it was not easy to find its description (please mention it also here).

Line 78: The same with "FECs"!

Line 75 onwards: I suggest being cautious when talking about "relationships". What about statistical significances of the correlations?

Line 114 onwards: Problematically low fGCM concentrations have also been found in other animal studies (e.g. Dorsey et al., 2010 or Pawluski et al., 2017), which I suggest to include in the discussion.

Line 132: "social factors" were only indirectly evaluated (only by group composition), or?

Line 240: I think that's the wrong citation for a validation of the corticosterone EIA (CJM006) here.

Watson et al (2013) investigated Asian elephants and performed a biological validation in this species. However, only a test for parallelism was performed in African elephant fecal samples, which is not enough to prove that the EIAs is suited. Anyway, the EIA has been validated for African elephants (ACTH challenge test: see Santymire et al., 2012). Please replace the citation. What was the precision (intra- and inter-assay CVs of pool samples) of the EIA and its sensitivity (ng/g feces)?

Line 264 (and 270): Did you only use the NDVI for seasonality, or also (Julian) date?

Line 270-272: Compared to the fGCM analysis, there is only one sentence about NDVI – any details?

Fig 2: ndvi needs to be capitalized: NDVI

Fig 3 (lower panel). Please replace "Glucocorticoids" with "Glucocorticoid metabolites (ng/g)" (legend of y-axis). Here, but also in all graphs of the suppl Fig 2.

I welcome graphs with individual fGCM levels (suppl Fig 2). What about repeatability (you address this in line 68, but give no data/results). Concomitant peaks in all individuals of a group may indicate some specific stressor occurring about 24 h before sampling (see also Fig 3). I suggest including a statement about it in the discussion.

Page 41/42: Lines help to track individuals, but are misleading when sampling intervals are long.

References cited above:

- Dorsey, C., Dennis, P., Guagnano, G., Wood, T., Brown, J.L. (2010): Decreased baseline fecal glucocorticoid concentrations associated with skin and oral lesions in Black rhinos (*Diceros bicornis*). *J. Zoo Wildlife Med.* 41, 616-625.
- Pawluski, J., Jago, P., Henry, S., Bruchet, A., Palme, R., Coste, C., Hausberger, M. (2017): Low plasma cortisol and fecal cortisol metabolite measures as indicators of compromised welfare in domestic horses (*Equus caballus*). *PLoS ONE* 12, 12, e0182257.
- Santymire, R.M., Freeman, E.W., Lonsdorf, E.V., Heintz, M.R., Armstrong, D.M. (2012): Using ACTH challenges to validate techniques for adrenocortical activity analysis in various African wildlife species. *Int. J. Anim. Vet. Adv.* 4, 99-108.

Reviewer #2 (Remarks to the Author):

This study examines fecal glucocorticoid concentrations in wild female African elephants, focusing specifically on the differences (or lack thereof) between animals who were orphaned before sexual maturity and animals who were not. The study leverages the strengths of the extensive long-term research done on the Samburu elephants. The physiological effects of maternal loss—a specific type of early life adversity that is known to have life-long consequences in a wide variety of species—is a topic that is of great interest to scientists in a large number of disciplines right now.

I have a number of suggestions that I hope will improve the clarity of the manuscript. One of the challenges that comes with journals like *Comms Bio*, where methods come after the results, is that sufficient information must be included in the introduction to make sure that the reader can follow the results. This is a bit of an art. As written, this introduction renders the results quite confusing, because it fails to clarify some key details that are 100% necessary to understanding the results.

First, the term 'orphan' conjures up images of young animals for most people. I was quite confused upon my initial reading as to how/why 'lactating' and 'pregnant' could possibly be predictors in the model, since I had assumed (as I think most people would) that these were young animals who had lost their mothers in the recent past. It requires quite a bit of hunting to figure out that they can be as old as 21, and that none of them lost their mothers more recently than 2 years before data collection took place. There need to be a few additional sentences in the introduction that make it clear that the subjects span quite a wide range of life stages...the youngest animal was 2, meaning it was likely still nursing (was this controlled for? could likely have a major impact on fecal GCs), while the oldest was 21, and therefore conceivably multiparous.

In this vein, I do have questions—not sure they are clearly answerable—about the biological relevance of lumping all these animals together. Was time since maternal loss considered as a predictor? I recognize that there is evidence of long-term effects of maternal loss on HPA axis function in e.g. humans, but a 4 year old elephant who lost its mother two years ago seems quite different than an 18 year old elephant who lost her mother 10 years ago. Similarly, would we really predict that an 18 year old animal who lost her mother 10 years ago (categorized as an orphan in this study) would have higher fGCMs than a similarly-aged animal who lost her mother 6 months ago (categorized as a non-orphan)? I am not necessarily sure that lumping them all together like this is helpful, though I recognize that it was almost certainly done because of the inevitable constraints of sample sizes with studies like this. However, it would be good if there was some additional discussion of the potential limitations and pitfalls of this approach.

Second, the results would be much easier to understand if they were written in biological terms, rather than statistical. 'The negative coefficient value for non-natal group also depicted a strong relationship' requires a lot of mental gymnastics for the reader. I would strongly prefer that this be phrased as an interpretable, biological result, e.g. 'Animals who were in their natal group had higher

fGCM concentrations than those who were not.' 'Animals had higher concentrations of fGCMs in years where there was greater fluctuation in available vegetation,' etc. NDVI was particularly confusing because it is not defined until the methods, but the problem with lack of translation to biological terms is not unique to this particular predictor variable.

Third, there are various places where the citations and background information feel incomplete and/or out of date. There has been quite a lot of work published in this general area on primates in the last couple of years. A probably incomplete list:

Girard-Buttoz, C., Tkaczynski, P. J., Samuni, L., Fedurek, P., Gomes, C., Löhrich, T., ... & Crockford, C. (2021). Early maternal loss leads to short-but not long-term effects on diurnal cortisol slopes in wild chimpanzees. *Elife*, 10, e64134.

Morrison, R. E., Eckardt, W., Colchero, F., Vecellio, V., & Stoinski, T. S. (2021). Social groups buffer maternal loss in mountain gorillas. *Elife*, 10, e62939.

Rosenbaum, S., Zeng, S., Campos, F. A., Gesquiere, L. R., Altmann, J., Alberts, S. C., ... & Archie, E. A. (2020). Social bonds do not mediate the relationship between early adversity and adult glucocorticoids in wild baboons. *Proceedings of the National Academy of Sciences*, 117(33), 20052-20062.

Campos, F. A., Archie, E. A., Gesquiere, L. R., Tung, J., Altmann, J., & Alberts, S. C. (2021). Glucocorticoid exposure predicts survival in female baboons. *Science advances*, 7(17), eabf6759.

The claims that '...our current understanding of maternal effects on adrenal activity and the potential for other relationships to offset stress responses associated with maternal loss has been derived entirely from captive studies, without insight from studies in wild individuals,' and 'our study was unique in investigating how maternal loss affects adrenal activity in wild orphans of a long-lived mammal species' just simply aren't true. The Girard-Buttoz et al. and Rosenbaum et al. studies above do exactly that (as does Morrison et al. if we are talking specifically about the potential for other relationships to buffer maternal loss), and frankly, the collective results paint a much more complicated picture than what is conveyed in this manuscript. Another example: when talking about the potential health effects of long-term glucocorticoid exposure, it seems much more appropriate to cite something like Campos et al. than it does a popular science book that was published nearly 30 years ago. I think it would be prudent to go back and do another literature search that focuses on, say, the last 5 years, to be sure that the info in the introduction and discussion accurately reflects the current state of the literature.

In addition to these bigger-picture suggestion, I list some specific questions and comments below. I hope the authors will find them useful as they revise their paper.

Throughout: so are animals who are defined as orphans or the non-orphan 'matches' they are compared to, also classified as adult caregivers if they are multiparous? Not clear from the description whether any of these 40 animals would themselves count as adult caregivers...it's not impossible that the older animals in the data set could be multiparous. Similar question about age mates; are the subjects themselves both subjects, and age mates for other animals?

Line 41: I find 'cognitively advanced' to be a rather strange, and undefined, term. 'Slow life history' or 'high encephalization quotient' (if you want to focus specifically on the brain) would be less ambiguous.

Line 78: What does 'strongylid FECs dropped out of the model run with fewer samples, suggesting no

correlation' mean? Correlation between what; strongylid infection and fecal GCMs? What is this actually telling us, in biological terms, getting back to the more general comment above?

Line 240: to be totally transparent, I think you should mention that the biological validation for this assay was only done in Asian elephants. The chances it would be different in African elephants is tiny, but it might strike some people as disingenuous to neglect to mention that.

Line 277: Is there anything in these models that attempts to deal with intra-individual correlation? I would rather see animal ID included as a fixed effect than number of samples/individual; I am guessing using both might not be possible due to high collinearity, but I could be wrong.

Figure 3: It is very helpful for readers if figures can stand on their own, without reading the text. Define NDVI in the figure caption.

Supplement, line 6: It's kind of interesting that you see a time-of-day effect of fGCM in zoo elephants, but this is not all that surprising in an animal with a long gut passage time. Many other studies have shown a lack of time of day effect in megafauna.

Supplementary Table 1: This table is really confusing. 'We then calculated the DIC for the global model minus combinations of variables without which the DIC decreased' is hard to parse (from lines 321-322 in the methods), so I'm still not clear on how exactly this subset of all the potential models was arrived at. Why is there no caption that provides the name of the variable the letter(s) in the 'variables' column represents? Exactly which column is the DIC scores? You can't call it one thing in the table description and another thing in the table itself. Again, there is a general lack of translation to biology here that is problematic. It assumes too much familiarity with the data and the model selection technique.

Supplementary table 3: how big is the sample size here? Give us the n, not just 'fewer samples.' Is this what you mean when you say 'sample set 2' in e.g. supplementary table 1, or is that referring to something different?

Supplementary figure 2: In the time series graphs, put dots when the actual samples were collected. I want to know whether these lines are being drawn using (e.g.) 20 data points, or 3.

Reviewer #3 (Remarks to the Author):

The authors have examined faecal glucocorticoid metabolites concentration in African elephant to understand physiological stress level with reference to social relationships in family members. They also examined the effect of maternal loss on young elephants (orphan) along with non-orphans. They also examined natal vs non-natal orphan's stress level. They didn't find any significant difference with these sets might be due to social bonding and family relationship helping them to overcome the loss of mother, unlike other mammalian species. This is very interesting result as a whole because the social relationship and family bonding are very important for group living animals. The data was analysed well and well written.

My only concern is that the study was carried almost 2yr after the orphaning the calf which might not reflect the stress as calf would have undergone stress full event immediately after loss the mother which is correct way of examining them. After two years calfs would have overcome stress because of family relationships.

Referee expertise:

Referee #1: glucocorticoids, stress assessment of wild mammals

Referee #2: physiological mediators of social behavior

Referee #3: elephants, physiological stress assessment

Reviewer #1 (Remarks to the Author):

The paper "Social support correlates with glucocorticoid concentrations in wild African elephant orphans", submitted by Parker and colleagues investigated the effect of maternal loss on fecal glucocorticoid metabolite (fGCM) concentrations in wild African elephants. In their interesting, longitudinal field study of well monitored African elephant groups in Kenya, the authors found lower fGCM levels in juveniles with more individuals in the core group, but no differences between orphans and non-orphans. Unexpectedly, orphans which had left their natal family had also lower fGCM concentrations.

I found it confusing that none of the stated (e.g. lines 5, and 7-8) and discussed (line 92 onwards) main findings can be explicitly found in the "Results" section, but only technical statements pointing to tables and figures. What about the statistical significance of those findings.

Thank you, we have revised the language in the results section to be more biological, which hopefully clarifies those results within the text. In terms of statistical significance, our stats were not frequentist and therefore did not involve traditional p-values. As suggested by the American Statistical Association, we focused on variables whose 95% confidence intervals did not overlap zero as showing a statistically significant correlation, and clarified this in the results (see line 97).

In addition, some findings of the abstract (e.g. line 7) seem to contradict the title.

We included "orphans" in the title to encompass what the study originally set out to test, even though we did not find the results we expected with respect to orphans versus non-orphans but rather that social support matters in more nuanced ways.

Nevertheless, I enjoyed reading and think the paper warrants publication.

Thank you!

However, these and other points need to be clarified by the authors (see also detailed comments below).

Below, please find specific comments and questions (ordered by appearance in the ms):

Line 11: I don't think "resilience" can be concluded from your data and from a mere correlational study.

We have modified this language, please see lines 15 – 16.

Line 13: “benefits of sociality”: I guess the other findings (lines 5-6) may underline this)

We have modified the wording to be more specific, especially given the new results after removing the very young nonorphan subjects from our analysis, as suggested by multiple reviewers. Please see lines 15 - 16.

Line 13: “and contributing to our understanding of why it evolved”. I suggest deleting this part. It would be similar true in case of opposite results and what do we understand here now better?

We have deleted as suggested.

Line 66: A minimum fGCM concentration of 10 pg/g seems very low. What was the detection limit of the method (in ng/g)? And by looking at the suppl Fig 2, this value seems to be an artefact (not reasonable – and there are a few other of those very low levels – please re-check). But most likely those were already removed (lines 292-294).

Please note that our measurements are in ng/g, not pg/g. Thank you for pointing this out though! We found a typo in the data sheet that we had not noticed, resulting in a value of 0.01 ng/g that should have been 111.25 ng/g. We have included the data sheet in supplementary materials (now linked at the end of the manuscript as they are uploaded to Harvard Dataverse). The fixed value is in row 263 of the excel sheet. The other low measure was 14.19 ng/g, which is very low but still possible, so we did not have a good reason for discarding it.

We inserted the assay sensitivity in lines 289 – 290; it was 0.14 ng/ml. We do not refer to a detection limit on a per gram basis because it is affected by a number of other factors, such as the dilution factor, weight of fecal powder used, and the individual sample extraction efficiency. So there is no single number we can refer to for this assay. However, if we assume a fecal weight of 0.1 g, dilution factor of 5, and extraction efficiency of 100%, then the lowest value could be 0.14 ng/g. However, that would not apply to all samples if they had different weights or dilution factors. We also do not use sample binding values outside of 20 – 80%, so we would not calculate a concentration for a sample binding at 90%. Thus, we hope it is okay to just list the assay sensitivity.

Line 72: NDVI. This abbreviation pops up here for the first time and it was not easy to find its description (please mention it also here).

Sorry, that was defined in the methods, which come after results in this journal. We revised so the definition is available the first time we use the abbreviation in the results, see line 111 – 112.

Line 78: The same with “FECs”!

Same as above, we have added an explanation for the abbreviation the first time “FECs” is used, see line 122.

Line 75 onwards: I suggest being cautious when talking about “relationships”. What about statistical significances of the correlations?

We have amended our wording throughout so that the word “relationship” is only used in the context of social relationships, instead being careful to focus on correlation.

Line 114 onwards: Problematically low fGCM concentrations have also been found in other animal studies (e.g. Dorsey et al., 2010 or Pawluski et al., 2017), which I suggest to include in the discussion.

Thank you, we have referred to these studies in our discussion of hypocortisolism, see lines 163 – 167.

Line 132: “social factors” were only indirectly evaluated (only by group composition), or?

Yes, and also orphan status and whether an orphan was with related individuals, because these have large social consequences. We have changed the word “factors” to “context” since this seems to portray our meaning better, see line 178.

Line 240: I think that’s the wrong citation for a validation of the corticosterone EIA (CJM006) here. Watson et al (2013) investigated Asian elephants and performed a biological validation in this species. However, only a test for parallelism was performed in African elephant fecal samples, which is not enough to prove that the EIAs is suited. Anyway, the EIA has been validated for African elephants (ACTH challenge test: see Santymire et al., 2012). Please replace the citation. What was the precision (intra- and inter-assay CVs of pool samples) of the EIA and its sensitivity (ng/g feces)?

We cited Watson et al. because they did a lot more work on the CJM006, but have added Santymire et al. 2012 as you have suggested since that was in African elephants. We have also added in the sensitivity and precision of the EIA in lines 289 – 290.

Line 264 (and 270): Did you only use the NDVI for seasonality, or also (Julian) date?

We only used NDVI. The timing of seasonal transitions in the study area are highly variable, making Julian date less informative. Previous research has shown seasonality as measured by primary productivity is biologically meaningful for elephants.

Line 270-272: Compared to the fGCM analysis, there is only one sentence about NDVI – any details?

We detail how we calculated NDVI (using the core area drawn from collar data, and how we interpreted mean versus standard deviation) in lines 310 – 316.

Fig 2: ndvi needs to be capitalized: NDVI

We fixed this, thank you.

Fig 3 (lower panel). Please replace “Glucocorticoids” with “Glucocorticoid metabolites (ng/g)” (legend of y-axis). Here, but also in all graphs of the suppl Fig 2.

We fixed this in Figure 3 and Supplementary Figure 2.

I welcome graphs with individual fGCM levels (suppl Fig 2). What about repeatability (you address

this in line 68, but give no data/results). Concomitant peaks in all individuals of a group may indicate some specific stressor occurring about 24 h before sampling (see also Fig 3). I suggest including a statement about it in the discussion.

The repeatability among the samples from a single individual was very low, 0.048 (CI .001, 0.105) as calculated in the R package rptR (citation below). The large influence of seasonality is likely the cause of concomitant peaks – it was a population-wide phenomenon that glucocorticoids fluctuated with available resources. We shy away from making repeatability statements because, given constant environmental conditions, we may have found an individual's concentrations to be very repeatable.

(Stoffel, M.A., Shinichi, N., & Schielzeth, H. (2017) rptR: Repeatability estimation and variance decomposition by generalized linear mixed-effects models. *Methods in Ecology and Evolution* 8(11): 1639-1644. doi: 10.1111/2041-210X.12797.)

Page 41/42: Lines help to track individuals, but are misleading when sampling intervals are long.

We added dots at each of the sampling points of Supplementary Figure 2 to help clarify where samples were taken (as well as in Figure 3). We also made each of the graphs in Supplementary Figure 2 much larger so you can adequately see what is happening, possible as the figure and its many graphs are now uploaded onto Harvard Dataverse.

References cited above:

Dorsey, C., Dennis, P., Guagnano, G., Wood, T., Brown, J.L. (2010): Decreased baseline fecal glucocorticoid concentrations associated with skin and oral lesions in Black rhinos (*Diceros bicornis*). *J. Zoo Wildlife Med.* 41, 616-625.

Pawluski, J., Jago, P., Henry, S., Bruchet, A., Palme, R., Coste, C., Hausberger, M. (2017): Low plasma cortisol and fecal cortisol metabolite measures as indicators of compromised welfare in domestic horses (*Equus caballus*). *PLoS ONE* 12, 12, e0182257.

Santymire, R.M., Freeman, E.W., Lonsdorf, E.V., Heintz, M.R., Armstrong, D.M. (2012): Using ACTH challenges to validate techniques for adrenocortical activity analysis in various African wildlife species. *Int. J. Anim. Vet. Adv.* 4, 99-108.

Reviewer #2 (Remarks to the Author):

This study examines fecal glucocorticoid concentrations in wild female African elephants, focusing specifically on the differences (or lack thereof) between animals who were orphaned before sexual maturity and animals who were not. The study leverages the strengths of the extensive long-term research done on the Samburu elephants. The physiological effects of maternal loss—a specific type of early life adversity that is known to have life-long consequences in a wide variety of species—is a topic that is of great interest to scientists in a large number of disciplines right now.

I have a number of suggestions that I hope will improve the clarity of the manuscript. One of the challenges that comes with journals like *Comms Bio*, where methods come after the results, is that sufficient information must be included in the introduction to make sure that the reader can follow the results. This is a bit of an art. As written, this introduction renders the results quite confusing,

because it fails to clarify some key details that are 100% necessary to understanding the results.

Thank you for pointing this out. Given the journal format, we have modified to include key details at the end of the Introduction and Results sections that before could only be found in the methods.

First, the term ‘orphan’ conjures up images of young animals for most people. I was quite confused upon my initial reading as to how/why ‘lactating’ and ‘pregnant’ could possibly be predictors in the model, since I had assumed (as I think most people would) that these were young animals who had lost their mothers in the recent past. It requires quite a bit of hunting to figure out that they can be as old as 21, and that none of them lost their mothers more recently than 2 years before data collection took place. There need to be a few additional sentences in the introduction that make it clear that the subjects span quite a wide range of life stages...the youngest animal was 2, meaning it was likely still nursing (was this controlled for? could likely have a major impact on fecal GCs), while the oldest was 21, and therefore conceivably multiparous.

We have added information concerning how long it has been since the orphans’ mothers died to the introduction (lines 76 – 78), as well as information on the subjects’ ages at the beginning of the results (lines 87 – 88). We have also removed the three youngest nonorphan subjects from analysis, given nursing could indeed have a major impact on fecal GC’s. For the older individuals, we included age as a covariate, and also accounted for reproductive status by including covariates for lactating and pregnancy. See below for more discussion on how we modified our analysis according to your comments.

In this vein, I do have questions—not sure they are clearly answerable—about the biological relevance of lumping all these animals together. Was time since maternal loss considered as a predictor? I recognize that there is evidence of long-term effects of maternal loss on HPA axis function in e.g. humans, but a 4 year old elephant who lost its mother two years ago seems quite different than an 18 year old elephant who lost her 10 years ago. Similarly, would we really predict that an 18 year old animal who lost her mother 10 years ago (categorized as an orphan in this study) would have higher fGCMs than a similarly-aged animal who lost her 6 months ago (categorized as a non-orphan)? I am not necessarily sure that lumping them all together like this is helpful, though I recognize that it was almost certainly done because of the inevitable constraints of sample sizes with studies like this. However, it would be good if there was some additional discussion of the potential limitations and pitfalls of this approach.

We have included a covariate for time since mother’s death in our modified analysis. We forewent model selection (see below) so that all variables remained in the model, therefore variation due to time since mother’s death was accounted for. We have also added throughout clarifying statements (e.g. lines 3 – 4, 74 – 78, and 135 – 136) that we were looking at lasting differences several years following the death of an individual’s mother. We state the mean and standard deviation of time since mother’s death in the introduction (with the range included in the methods), and the range, mean and standard deviation of age at the beginning of the results so readers are aware of that context from the outset.

Second, the results would be much easier to understand if they were written in biological terms, rather than statistical. ‘The negative coefficient value for non-natal group also depicted a strong relationship’ requires a lot of mental gymnastics for the reader. I would strongly prefer that this be phrased as an interpretable, biological result, e.g. ‘Animals who were in their natal group had higher fGCM concentrations than those who were not.’ ‘Animals had higher concentrations of fGCMs in

years where there was greater fluctuation in available vegetation,' etc. NDVI was particularly confusing because it is not defined until the methods, but the problem with lack of translation to biological terms is not unique to this particular predictor variable.

We have adjusted the language in the results to be more explicit, but also maintained some of the statistical language that we build on in the discussion. We give a definition of NDVI in the results (see line 110) and delve into the meaning of the NDVI correlations in the discussion.

Third, there are various places where the citations and background information feel incomplete and/or out of date. There has been quite a lot of work published in this general area on primates in the last couple of years. A probably incomplete list:

Girard-Buttoz, C., Tkaczynski, P. J., Samuni, L., Fedurek, P., Gomes, C., Löhrich, T., ... & Crockford, C. (2021). Early maternal loss leads to short-but not long-term effects on diurnal cortisol slopes in wild chimpanzees. *Elife*, 10, e64134.

Morrison, R. E., Eckardt, W., Colchero, F., Vecellio, V., & Stoinski, T. S. (2021). Social groups buffer maternal loss in mountain gorillas. *Elife*, 10, e62939.

Rosenbaum, S., Zeng, S., Campos, F. A., Gesquiere, L. R., Altmann, J., Alberts, S. C., ... & Archie, E. A. (2020). Social bonds do not mediate the relationship between early adversity and adult glucocorticoids in wild baboons. *Proceedings of the National Academy of Sciences*, 117(33), 20052-20062.

Campos, F. A., Archie, E. A., Gesquiere, L. R., Tung, J., Altmann, J., & Alberts, S. C. (2021). Glucocorticoid exposure predicts survival in female baboons. *Science advances*, 7(17), eabf6759.

The claims that '...our current understanding of maternal effects on adrenal activity and the potential for other relationships to offset stress responses associated with maternal loss has been derived entirely from captive studies, without insight from studies in wild individuals,' and 'our study was unique in investigating how maternal loss affects adrenal activity in wild orphans of a long-lived mammal species' just simply aren't true. The Girard-Buttoz et al. and Rosenbaum et al. studies above do exactly that (as does Morrison et al. if we are talking specifically about the potential for other relationships to buffer maternal loss), and frankly, the collective results paint a much more complicated picture than what is conveyed in this manuscript. Another example: when talking about the potential health effects of long-term glucocorticoid exposure, it seems much more appropriate to cite something like Campos et al. than it does a popular science book that was published nearly 30 years ago. I think it would be prudent to go back and do another literature search that focuses on, say, the last 5 years, to be sure that the info in the introduction and discussion accurately reflects the current state of the literature.

We have updated references and wording in the introduction to reflect the Girard-Buttoz et al. and Rosenbaum et al. studies, thank you, see lines 43 – 53. We were aware of the Morrison et al. study, but that does not deal with stress hormone measurements. We added the Campos et al. reference to line 35.

In addition to these bigger-picture suggestion, I list some specific questions and comments below. I hope the authors will find them useful as they revise their paper.

Throughout: so are animals who are defined as orphans or the non-orphan ‘matches’ they are compared to, also classified as adult caregivers if they are multiparous? Not clear from the description whether any of these 40 animals would themselves count as adult caregivers...it’s not impossible that the older animals in the data set could be multiparous. Similar question about age mates; are the subjects themselves both subjects, and age mates for other animals?

None of the subjects were multiparous when sampling began, therefore we did not include them in the adult caregiver category because we included multiparous in our definition of “adult caregiver” for the purposes of our analysis. (Primiparous females are still quite young and inexperienced, and therefore likely do not offer the same type of social support as older females.) And yes, subjects counted as age mates for one another when relevant.

Line 41: I find ‘cognitively advanced’ to be a rather strange, and undefined, term. ‘Slow life history’ or ‘high encephalization quotient’ (if you want to focus specifically on the brain) would be less ambiguous.

We changed to “a long-lived, highly social species with a high encephalization quotient that indicates advanced cognitive abilities”, see lines 54 – 56.

Line 78: What does ‘strongylid FECs dropped out of the model run with fewer samples, suggesting no correlation’ mean? Correlation between what; strongylid infection and fecal GCMs? What is this actually telling us, in biological terms, getting back to the more general comment above?

We have modified this sentence to be clearer, see lines 124 – 125. The finding is also discussed in the supplementary discussion.

Line 240: to be totally transparent, I think you should mention that the biological validation for this assay was only done in Asian elephants. The chances it would be different in African elephants is tiny, but it might strike some people as disingenuous to neglect to mention that.

We cited Watson et al. because they did a lot more work on the CJM006, but the method we used has also been validated for African elephants and we have added that citation (Santymire et al. 2012) to reflect this.

Line 277: Is there anything in these models that attempts to deal with intra-individual correlation? I would rather see animal ID included as a fixed effect than number of samples/individual; I am guessing using both might not be possible due to high collinearity, but I could be wrong.

Yes, the hierarchical model is robustly designed so that an individual’s samples are grouped together under a common intercept estimated from the average fGCM concentration across all of her samples. Each level of the model informs the other. We clarified this in lines 356 – 358.

Figure 3: It is very helpful for readers if figures can stand on their own, without reading the text. Define NDVI in the figure caption.

We added an explanation of NDVI to all relevant figures and tables.

Supplement, line 6: It’s kind of interesting that you see a time-of-day effect of fGCM in zoo

elephants, but this is not all that surprising in an animal with a long gut passage time. Many other studies have shown a lack of time of day effect in megafauna.

The zoo study used urinary samples, perhaps that had an effect? We were looking for the fGCM to reflect changes in glucocorticoid secretion throughout the day, which I would think they would if there were fluctuations even with a long gut passage time, the time would just need to be adjusted to account for passage. I am having trouble locating the studies you mention using Web of Science, perhaps because a lack of effect is less likely to be included in titles/abstracts.

Supplementary Table 1: This table is really confusing. 'We then calculated the DIC for the global model minus combinations of variables without which the DIC decreased' is hard to parse (from lines 321-322 in the methods), so I'm still not clear on how exactly this subset of all the potential models was arrived at. Why is there no caption that provides the name of the variable the letter(s) in the 'variables' column represents? Exactly which column is the DIC scores? You can't call it one thing in the table description and another thing in the table itself. Again, there is a general lack of translation to biology here that is problematic. It assumes too much familiarity with the data and the model selection technique.

We decided to make inference from a single model (the global model), which is appropriate in this case because each variable is present for a reason rooted in literature and/or with an objective toward testing our main questions. This decision removes the need for concern over model selection, and we have removed the Supplementary Table 1 of the prior submitted manuscript.

Supplementary table 3: how big is the sample size here? Give us the n, not just 'fewer samples.' Is this what you mean when you say 'sample set 2' in e.g. supplementary table 1, or is that referring to something different?

We have included the sample size in the caption (n = 446 samples from 37 elephants). This table is now Supplementary Table 1. It is linked at the end of the manuscript.

Supplementary figure 2: In the time series graphs, put dots when the actual samples were collected. I want to know whether these lines are being drawn using (e.g.) 20 data points, or 3.

We have inserted dots at sample collection points, see the link to Supplementary Figure 2 on Harvard Dataverse included at the end of the manuscript. (The graphs are also much larger now so they are easier to view.)

Reviewer #3 (Remarks to the Author):

The authors have examined faecal glucocorticoid metabolites concentration in African elephant to understand physiological stress level with reference to social relationships in family members. They also examined the effect of maternal loss on young elephants (orphan) along with non-orphans. They also examined natal vs non-natal orphan's stress level. They didn't find any significant difference with these sets might be due to social bonding and family relationship helping them to overcome the loss of mother, unlike other mammalian species. This is a very interesting result as a whole because the social relationship and family bonding are very important for group living animals. The data was analysed well and well written.

My only concern is that the study was carried almost 2yr after the orphaning the calf which might not reflect the stress as calf would have undergone stress full event immediately after loss the mother which is correct way of examining them. After two years calfs would have overcame stress because of family relationships.

Thank you for pointing this out. We have added lines to the last paragraph of the introduction clarifying for readers that it had been a long time since the orphans' mothers had died, and therefore we were looking for more lasting effects in orphans who manage to survive, see lines 76 – 78. Additionally, we have redone the analysis, adding a covariate of years since mother's death to account for variation due to the time since an orphan's mother had died (nonorphans were assigned a value of zero).

REVIEWERS' COMMENTS:

Reviewer #1 (Remarks to the Author):

Thanks to the author for significantly improving their ms. I'm happy with the revision, and only have a few, minor things left. I don't need to see the ms again after correction.

Line 122: I think it's better to have FECs for "fecal egg counts", instead of FEC's. Why should a possessive apostrophe make sense?

Line 337: Please add "metabolites" to "fecal glucocorticoid" – there are no glucocorticoids excreted in the feces, but only their metabolites.

"We added dots at each of the sampling points of Supplementary Figure 2 to help clarify where samples were taken (as well as in Figure 3)." I suggest using different symbols (not only black dots) for the different animals. That would strongly clarify the figs.

Reviewer #2 (Remarks to the Author):

Overall, I am relatively happy with the revisions that the authors made—thank you for taking the critiques of myself and the other reviewers seriously. The revision has clarified many of the questions I had the first time around. However, the increased clarity has also helped to highlight a remaining issue. While the data in the paper may be consistent with social buffering, it is tangential evidence at best, and the way social buffering is discussed needs to reflect this. There are two pieces to this: first, fGCMs and stress are not synonymous, and there are various places in the manuscript where the two get conflated. To truly demonstrate stress, one would need to demonstrate some physiological or behavioral symptom besides fGCMs (say weight loss or depressive behavior, for example) in conjunction with high fGCMs. Second, to truly demonstrate social buffering against stress, one would need to show that animals who had more social support were less likely to have high fGCMs and the other symptom(s). This paper shows some relationships between fecal GCs and social and ecological environment, which is interesting and important, but the claims it is making are too specific and go beyond what the data can actually speak to.

Furthermore, there is an unacknowledged inconsistency in interpretation. While it appears in various forms throughout, it is most succinctly stated in this sentence, from lines 175-177:

"Assuming social buffering from family members lowers GCs of orphans after an initial rise, hypocortisolism in non-natal elephant orphans is a plausible explanation for our results and indicates prolonged costs of their social situation."

First, as discussed above, this is requiring a lot of assumptions the current data do not actually speak to. But second, even if what family members are doing is buffering against hypercortisolism, then it does not necessarily follow that hypocortisolism is a result of lack of social buffering (and I am also rather skeptical that we can diagnose hypocortisolism here, without a lot more data on what the normal range for elephants is). It is also a very big leap to claim that this is evidence of a long-term cost, without any kind of data that demonstrate a cost. There are no fitness, health, or longevity data in this paper. If these were baboons, per the Campos et al. paper that is now cited, the animals with ostensible "hypocortisolism" would be expected to live meaningfully longer lives than the animals with high cort, and longevity is a very important predictor of lifetime fitness in many long-lived mammals.

I understand the temptation to over-interpret data, but the degree of over-interpretation that is going on here is, to my mind at least, unacceptable. The good news is that I think this problem can be fixed with some relatively minor wording tweaks that make it clearer what is speculation, and that outline what additional evidence we would need in order to clearly demonstrate 'stress' (not just elevated GC levels, which are not synonymous with stress), and to demonstrate social buffering.

Reviewer #3 (Remarks to the Author):

All my comments have been addressed.

REVIEWERS' COMMENTS:

Following these responses, there is a version of the manuscript with highlighted changes made in response to the reviewers' comments.

Reviewer #1 (Remarks to the Author):

Thanks to the author for significantly improving their ms. I'm happy with the revision, and only have a few, minor things left. I don't need to see the ms again after correction.

Line 122: I think it's better to have FECs for "fecal egg counts", instead of FEC's. Why should a possessive apostrophe make sense?

Thank you for pointing this out, we have removed the apostrophe.

Line 337: Please add "metabolites" to "fecal glucocorticoid" – there are no glucocorticoids excreted in the feces, but only their metabolites.

We amended to say fecal glucocorticoid metabolites.

"We added dots at each of the sampling points of Supplementary Figure 2 to help clarify where samples were taken (as well as in Figure 3)." I suggest using different symbols (not only black dots) for the different animals. That would strongly clarify the figs.

Thank you, this comment greatly improved these figures. We tried shapes but found circles with different color fill to be even clearer.

Reviewer #2 (Remarks to the Author):

Overall, I am relatively happy with the revisions that the authors made—thank you for taking the critiques of myself and the other reviewers seriously. The revision has clarified many of the questions I had the first time around. However, the increased clarity has also helped to highlight a remaining issue. While the data in the paper may be consistent with social buffering, it is tangential evidence at best, and the way social buffering is discussed needs to reflect this. There are two pieces to this: first, fGCMs and stress are not synonymous, and there are various places in the manuscript where the two get conflated. To truly demonstrate stress, one would need to demonstrate some physiological or behavioral symptom besides fGCMs (say weight loss or depressive behavior, for example) in conjunction with high fGCMs.

Thank you. We have amended to discuss only fGCM / GC concentrations and the stress **response**, as this describes a physiological process, as opposed to making any assertions about stress itself. For example, lines 144 – 145 now say "...and one reason for strengthening

such social bonds may be that they reduce GC concentrations” as opposed to “...and one reason for strengthening such social bonds may be that they reduce stress.” For other instances, please see the highlighted changes in the revision below.

Second, to truly demonstrate social buffering against stress, one would need to show that animals who had more social support were less likely to have high fGCMs and the other symptom(s). This paper shows some relationships between fecal GCs and social and ecological environment, which is interesting and important, but the claims it is making are too specific and go beyond what the data can actually speak to.

Thank you again, we have amended, for example by explicitly pointing out the correlational nature of the study (see lines 140 – 142, “However our study was correlational, therefore we cannot assert with certainty that the presence of age mates attenuated the stress response and caused the observed lower fGCM concentrations.”. We also clarified when we are speculating (e.g. up front in the abstract, line 8 says ...“which we speculate may be due to the development of hypocortisolism...”). Please see other highlighted areas for further examples of where we reworded to be more careful in our interpretations.

Furthermore, there is an unacknowledged inconsistency in interpretation. While it appears in various forms throughout, it is most succinctly stated in this sentence, from lines 175-177:

“Assuming social buffering from family members lowers GCs of orphans after an initial rise, hypocortisolism in non-natal elephant orphans is a plausible explanation for our results and indicates prolonged costs of their social situation.”

First, as discussed above, this is requiring a lot of assumptions the current data do not actually speak to. But second, even if what family members are doing is buffering against hypercortisolism, then it does not necessarily follow that hypocortisolism is a result of lack of social buffering (and I am also rather skeptical that we can diagnose hypocortisolism here, without a lot more data on what the normal range for elephants is). It is also a very big leap to claim that this is evidence of a long-term cost, without any kind of data that demonstrate a cost. There are no fitness, health, or longevity data in this paper. If these were baboons, per the Campos et al. paper that is now cited, the animals with ostensible “hypocortisolism” would be expected to live meaningfully longer lives than the animals with high cort, and longevity is a very important predictor of lifetime fitness in many long-lived mammals.

We have removed all direct suggestions that the lower concentrations observed in non-natal orphans may be associated with costs. We agree that we cannot diagnose hypocortisolism and we did not intend to write as though that is the case. However, given what we know from studies in the same system about the difficult social situation of orphan elephants (and non-natal orphan elephants in particular), that weaned orphan elephants in

general suffer lower survival which has unidentified physiological causes, that surviving orphan elephants are shorter than nonorphans (a study in review) and that multiple studies in other mammal species have shown hypocortisolism to result from a loss of maternal care, it is a plausible explanation for what we observed in non-natal orphans and one that deserves discussion. We tried to be clearer that this would involve an initial prolonged increase in GC secretion before downregulation of adrenals would occur, e.g. lines 170 – 172 now read, "**Supposing** the absence of family initially causes a prolonged rise in GC secretion for non-natal orphans, we are **potentially** observing the results of an eventual downregulation of the adrenals over time given we sampled several years after the death of orphans' mothers."

I understand the temptation to over-interpret data, but the degree of over-interpretation that is going on here is, to my mind at least, unacceptable. The good news is that I think this problem can be fixed with some relatively minor wording tweaks that make it clearer what is speculation, and that outline what additional evidence we would need in order to clearly demonstrate 'stress' (not just elevated GC levels, which are not synonymous with stress), and to demonstrate social buffering.

We feel we have been thorough in making the suggested tweaks and hope you will agree. Thank you for improving the rigor of the manuscript.

Reviewer #3 (Remarks to the Author):

All my comments have been addressed.